# Personnel Detection in Dark Aquatic Environments Based on Infrared Thermal Imaging Technology and an Improved YOLOv5s Model

**DOI:** 10.3390/s24113321

**Published:** 2024-05-23

**Authors:** Liang Cheng, Yunze He, Yankai Mao, Zhenkang Liu, Xiangzhao Dang, Yilong Dong, Liangliang Wu

**Affiliations:** 1School of Ocean Engineering, Jiangsu Ocean University, Lianyungang 222005, China; lance.cheng@yunzhou-tech.com; 2Zhuhai Yunzhou Intelligence Technology Co., Ltd., Zhuhai 519085, China; 3College of Electrical and Information Engineering, Hunan University, Changsha 410082, China; maom0406@163.com (Y.M.); hnulzk@163.com (Z.L.); dxz2825374263@163.com (X.D.); yilong.dong@student.manchester.ac.uk (Y.D.); liangl_w1010@163.com (L.W.)

**Keywords:** infrared thermal image, object detection, intelligent rescue, YOLO

## Abstract

This study presents a novel method for the nighttime detection of waterborne individuals using an enhanced YOLOv5s algorithm tailored for infrared thermal imaging. To address the unique challenges of nighttime water rescue operations, we have constructed a specialized dataset comprising 5736 thermal images collected from diverse aquatic environments. This dataset was further expanded through synthetic image generation using CycleGAN and a newly developed color gamut transformation technique, which significantly improves the data variance and model training effectiveness. Furthermore, we integrated the Convolutional Block Attention Module (CBAM) at the end of the last encoder’s feedforward network. This integration maximizes the utilization of channel and spatial information to capture more intricate details in the feature maps. To decrease the computational demands of the network while maintaining model accuracy, Ghost convolution was employed, thereby boosting the inference speed as much as possible. Additionally, we applied hyperparameter evolution to refine the training parameters. The improved algorithm achieved an average detection accuracy of 85.49% on our proprietary dataset, significantly outperforming its predecessor, with a prediction speed of 23.51 FPS. The experimental outcomes demonstrate the proposed solution’s high recognition capabilities and robustness, fulfilling the demands of intelligent lifesaving missions.

## 1. Introduction

Water rescue missions in dark environments pose a significant global challenge due to the complex surroundings and extremely low visibility that hinder the search for human subjects. Currently, infrared thermal imaging is predominantly utilized to detect pedestrians and vehicles on streets, with less focus on aquatic scenarios. Furthermore, water scene monitoring primarily depends on visible halo-scanning cameras stationed onshore, analyzing video footage manually. This approach is not only expensive but also ineffective for nighttime detection. Therefore, developing an accurate, fast, and intelligent human target detection technology for nighttime aquatic environments holds considerable practical importance. Recent advancements in object detection technology have expanded from PC-based systems to embedded and mobile platforms, with model recognition capabilities surpassing human performance. This progress underscores the theoretical and practical significance of exploring intelligent object detection and rescue methods for nighttime water environments.

Our research focuses on an intelligent surface object detection method tailored for dark conditions, exemplified by the “Dolphin I” surface lifesaving robot. This method aims to support search and rescue operations using “Dolphin I”, necessitating high accuracy and real-time capabilities. As illustrated in Figure 1, the “Dolphin I” is a model of a small aquatic lifesaving robot developed by the Zhuhai Yunzhou Intelligent Technology Company, designed for use in various drowning rescue scenarios such as swimming pools, reservoirs, rivers, and beaches.

It is well acknowledged that aquatic environments at night suffer from an extreme lack of light. Traditional object detection methods, which rely on visible-light imagery, perform poorly and struggle to accurately detect objects in water. In contrast, intelligent detection methods utilizing infrared thermal imaging technology [1] can map the temperature distribution of objects, offering the significant benefit of being unaffected by light conditions. Hence, we explore a detection method that harnesses the light-independent advantages of thermal imaging technology combined with deep learning-based object detection algorithms to address detection in dark environments. However, this research process encounters several challenges:(1)Practical aquatic scenes often contain numerous small objects that overlap, especially when the resolution of thermal images is lower than that of visible-light images, making objects smaller than 10 × 10 pixels. Additionally, human subjects often overlap significantly at vessel junctions, posing a significant challenge to the algorithm.(2)This technology’s application in outdoor aquatic environments is complicated by the variability of seasons, weather, and regional climate differences, which all impact thermal imaging.(3)Infrared cameras depend on temperature differences for image capture, leading to relatively low contrast in thermal images when the environmental temperature approximates that of the objects. Moreover, the varying emissivities of different materials can cause diverse energy radiation from surfaces, affecting imaging results. These issues can lead to misrecognition or incorrect categorization across different environments and subjects.(4)The practical deployment of the model also presents challenges, particularly with respect to maintaining model functionality. Our current plan involves using the NVIDIA Jetson Xavier platform, which requires model weights to be under 180 MB and an inference speed of over 25 frames per second (FPS).

To address these challenges, we propose an intelligent detection method for surface objects based on thermal imaging technology. Our contributions include the following:(1)This study employed traditional image enhancement methods such as horizontal flipping, cropping, and scaling, as well as restricted histogram equalization and mosaic enhancement methods to enhance the images, thereby improving the recognition accuracy of small overlapping objects for the algorithm.(2)This work created a thermal-image dataset for surface objects, enriched with various scenes including normal surface objects, drowning humans, and rescued individuals, thereby addressing the limitations of existing datasets for intelligent lifesaving missions. And the dataset was collected under various weather and water conditions during the acquisition process.(3)This work developed a thermal imaging color–space-transformation data augmentation technique. This method significantly enhances the distinguishability between objects and the environment by altering the RGB pixel values of detected targets and the background, thus eliminating the influence of imaging issues on detection performance.(4)This study developed a lightweight thermal imaging water object detection network, named IWT-YOLO (Infrared Water Target–YOLO), tailored to the needs of water object detection missions and the characteristics of thermal images. This model demonstrated superior performance on our self-constructed thermal-image dataset, accurately recognizing boats, intelligent lifesaving robots, humans, and humans in different conditions such as drowning, swimming, dabbling, being rescued, on board, and on the coast.

The remainder of this paper is structured as follows: Section 2 reviews recent research in this field, comparing the performance of various models and the rationale behind the model selection. Section 3 introduces the thermal imaging aquatic-object dataset created by our team. Section 4 discusses enhancements to the YOLOv5s algorithm. Section 5 presents the experimental results and the final section concludes the paper.

## 2. Related Work

Water object detection and thermal imaging object detection technologies have emerged as significant areas of research. Initially, visual perception technology was limited by the technological capabilities of its time, relying heavily on methods like wavelet transformation [2], manual feature extraction, and wave filters [3], which struggled with detecting and tracking complex targets. Q Li and colleagues [4] developed a method utilizing a region proposal network alongside a hierarchical section filter layer, leveraging deep features for detecting vessels in ultra-high-resolution optical remote sensing images, with sizes ranging from dozens to thousands of pixels. Y Peng [5] and team proposed a water object detection approach based on the Faster R-CNN algorithm, tailored for complex aquatic environments, which successfully identified and classified six common surface objects such as vessels and people. Liang Cheng [6] and associates introduced an integrated water object recognition system for unmanned vessels, employing an enhanced YOLOv4 algorithm [7], which demonstrated high accuracy and speed on the Nvidia Jetson AGX Xavier edge computing platform. Although most water object detection research has focused on visible-light images, the findings still offer valuable insights for our thermal imaging-based studies.

In terms of thermal image processing and recognition, significant strides were made as early as the last century. R. Strickland [8] applied wavelet transform filtering in 1997 to extract vessel contours from thermal images. Withagen [9] and colleagues conducted classification studies using vessel images from forward-looking infrared cameras in 1999. More recently, the advancements in infrared thermal imaging and computer vision have led to the application of deep learning techniques to thermal imagery. Fatih Altay [10] introduced an enhanced RetinaNet object detection network, demonstrating impressive accuracy on thermal-image datasets like Kaist and OSU. Masayuki Shimoda [11] designed an FPGA implementation of an object detector based on the sparse YOLOv2 [12] algorithm, capable of processing thermal and visible images simultaneously, achieving an inference speed of 22.4 FPS. Chi-Chia Sun [13] and team devised an algorithm for detecting fast-moving objects using complementary depth image and color information, effectively eliminating background noise. Wanzeng Kong [14] proposed Yolov3-DPFin, an improved real-time detection algorithm using a dual-path network (DPN) module and a fusion transition module for efficient feature extraction and enhanced multi-scale prediction. Feng Hong [15] and colleagues improved network precision and generalizability by harnessing deep residual network structures for high-level semantic target information. Ganbayar Batchuluun [16] developed a thermal image restoration method using deep learning, achieving state-of-the-art results in super-resolution reconstruction and deblurring. Mate Kristo [17] conducted research in thermal image object detection for security, utilizing YOLOv3 to detect illegal nighttime activities with remarkable efficacy. MA Farooq [18] produced a large-scale thermal image dataset and improved the YOLOv5s network, showcasing superior performance in thermal object detection for intelligent vehicle safety systems. While much of the thermal image object detection research has concentrated on security, pedestrian, and vehicle detection, studies focused on aquatic environments remain scarce, underscoring the significance of our work in water object detection with thermal images.

## 3. Construction of the Thermal-Image Aquatic Object Dataset

This paper explores the application contexts of object detection technology to furnish a foundation for decision making regarding the nighttime detection of water objects and the smart lifesaving capabilities of unmanned vessels. The effectiveness of object detection models that utilize deep learning hinges significantly on both the volume and the quality of the datasets employed. The impact of environmental variables such as weather on the performance of these models can be significantly mitigated by training the models using a broad array of thermal imagery data of aquatic environments across various regions, seasons, and climatic conditions.

### 3.1. Infrared Water Target Datasets

Given the lack of thermal imaging datasets for individuals in nocturnal aquatic environments, and in alignment with the principles of Infrared Machine Vision (IRMV) founded on infrared imaging technology as developed by Yunze He [19] and colleagues, our team has deliberated on the utilization of deep learning within the IRMV framework. Furthermore, we have conducted on-site examinations of aquatic ecosystems across various regions in China, incorporating both camera and thermal data into our research. Meanwhile, to mitigate the impact of environmental conditions on model performance, our data collection process lasted for 10 days, during which we gathered datasets under various weather conditions as extensively as possible. However, due to the high time and economic costs associated with data collection, our dataset was not systematically categorized according to different environmental conditions (such as lighting, water quality, weather, etc.), nor did it include certain specific scenarios, such as during rain, when there are strong winds with visible waves on the water surface, or when the ambient temperature is close to that of the objects.

For data collection, we employed the MAG62 and H20T thermal imagers, both of which are capable of saving the acquired data as images and videos. The specifications of these instruments are detailed in Table 1. Our team gathered images of individuals in natural water environments at night from various angles, utilizing drones and capturing footage from vessels or the shore. Additionally, this study generated a subset of thermal imaging data using Generative Adversarial Networks (GANs), resulting in over 70,000 original images. After a process of elimination, screening, and labeling, we constructed a comprehensive thermal imaging dataset of aquatic subjects. This dataset includes valuable samples such as scuffles in shallow-water areas, nighttime swimming in natural bodies of water, and simulated drowning struggles performed by expert winter swimmers. These samples provide significant data support for research in thermal imaging for water security. Selected data samples are presented in Figure 2.

The thermal imaging dataset for water objects comprises 4186 thermal images. Tailored to specific requirements, it categorizes into eight distinct classes: people drowning (drowning), people playing in the water (waterplaying), individuals being rescued by unmanned vessels (waterhelping), swimming (swimming), people on the shore (person_shore), people on board (person_boat), boats (boat), and the lifesaving robot “Dolphin I” (dolphin I). This dataset spans a variety of scenes including natural waters, coastlines, and artificial lake parks, ensuring a rich and diverse data collection. Additionally, the IR-YZ dataset features objects of varying sizes, ranging from small objects less than 10 × 10 pixels at long distances to larger, close-range objects. The dataset’s structure is detailed in Table 2.

### 3.2. Data Augmentation Method

The enhancement of model performance can be further achieved by applying appropriate data augmentation techniques to comprehensively collected datasets. In this study, conventional data augmentation methods such as horizontal/vertical flipping, rotation/reflection, and random scaling were utilized. Meanwhile, we employed the Contrast-Limited Adaptive Histogram Equalization (CLAHE) method, which enhances image contrast while avoiding the introduction of noise and unnatural effects. Additionally, we employed the mosaic data augmentation technique proposed by Glen Jocher, which diverges from traditional methods by employing multiple images (typically four) for random cropping, scaling, rotating, etc., and subsequently combining them into a single comprehensive image. The mosaic enhancement technique tends to reduce the area of larger objects while providing more training opportunities for smaller objects, thereby facilitating the network’s ability to recognize smaller objects. JY Zhu and colleagues introduced the CycleGAN [8] model, which facilitates the translation of unpaired images. Given the challenge of collecting thermal images of water objects online compared to the abundance of visible-light images, we leveraged the internet’s vast resources of visible-light images to explore thermal image object detection methods through a robust image translation [20] model. Thus, alongside the thermal images, a subset of visible-light images was also collected. A small-scale training dataset was created using thermal and visible-light images from similar scenes to evaluate the effectiveness of translating visible-light images into thermal images. A total of 1209 visible-light images and 1209 thermal images featuring the required water objects and shore-based objects were selected for training. After 200 training cycles, the CycleGAN model began to produce some distinctly clear thermal images. By making extensive use of visible-light images for image generation and selecting higher-quality images for thermal image dataset expansion, 78 superior thermal images were chosen to augment the training dataset. Some of the generated images are showcased in Figure 3.

Thermal images can be displayed in a vast array of pseudo-color formats. Preliminary research indicates that the method of color toning significantly affects the outcomes of object detection models. If all thermal images within a dataset are processed with a single toning method, the trained model struggles to recognize thermal images processed with different toning methods. As demonstrated in Figure 4, a model trained exclusively on thermal images with “white heat” toning performs poorly when applied to the same thermal images with “iron red” toning.

The color gamut transformation data augmentation technique for thermal images adjusts the RGB values of the thermal images or alters their gamma values, saturation, and brightness, among other parameters. The aim of this technique is to enhance the model’s ability to generalize across thermal images processed with various toning methods, as illustrated in Figure 5.

To prevent overfitting, we randomly chose only 20% of the images for color gamut transformation prior to training. Both the original and transformed images were then imported into the model for mixed training. As depicted in Figure 6, the model was capable of accurately recognizing thermal images processed with various toning methods after undergoing this training regimen.

## 4. Improvement of Object Detection Algorithm

Currently, deep learning-based object detection frameworks predominantly utilize either a two-step or a one-step approach. The two-step method, exemplified by Faster R-CNN [21], offers a high detection accuracy but suffers from slow inference speeds. On the other hand, the one-step method, represented by algorithms such as SSD [9] and YOLO, boasts significantly faster inference speeds. When tackling the task of water object detection using thermal images, prioritizing algorithm efficiency and computational resource conservation is essential. Consequently, we experimented with various networks, including SSD [22], YOLOv3 [23], and YOLOv5. The comparative performance of each model, using the thermal-image water object dataset, is detailed in Table 3.

Recognition speed is quantified by frames per second (FPS), while recognition capability is assessed using the mean average precision (mAP). A predicted bounding box is considered for evaluation if its intersection over union (IoU) with the ground-truth bounding box exceeds 0.5.

Taking into account various factors such as detection accuracy, inference speed, and the frequency of training, YOLOv5s was selected as the foundational network. YOLOv5s is a lightweight model within the YOLOv5 model series, characterized by a trained model size of merely 14 MB, which facilitates swift inference and rapid deployment. However, given that thermal images are significantly influenced by ambient temperature changes, the model requires robust feature extraction capabilities to recognize water objects amidst substantial background noise. The following are the enhancement strategies for YOLOv5s.

### 4.1. Convolutional Block Attention Module (CBAM)

The CBAM [24] (Convolutional Block Attention Module) integrates spatial and channel attention mechanisms, offering superior performance compared to the SENet attention mechanism, which focuses solely on channels. This is illustrated in Figure 7.

Being a lightweight and versatile module, the CBAM can be effortlessly incorporated into any CNN architecture with negligible computational overhead, allowing for end-to-end training alongside the base CNN. In this work, we enhance the feature extraction capabilities of the YOLOv5s backbone network by integrating the CBAM into it.

### 4.2. Exploiting Ghost Convolution

Ghost convolution, originating from GhostNet [25], is a convolutional module designed to reduce the computational load. Its approach divides the traditional convolution into two stages to lessen the network’s computational demands. Initially, traditional convolution with reduced computational intensity is used to generate feature maps with fewer channels. Subsequently, new feature maps are created based on these initial maps using a computationally inexpensive operation (depth-wise convolution), and the two sets of feature maps are concatenated to form the final output. With comparable computational resources, Ghost convolution demonstrates superior performance over the MobileNet-V3 network. In this work, selected convolutional layers of YOLOv5s were replaced with Ghost convolutional layers, aiming to maximize the inference speed without compromising model accuracy. The concept of Ghost convolution is depicted in Figure 8.

The enhanced network architecture is illustrated in Figure 9. This refined model demonstrates improved performance in both detection accuracy and inference speed. The detailed impacts of these enhancements will be thoroughly examined in the experimental results section.

### 4.3. Hyperparameter Evolution

Beyond the data preprocessing scheme, network architecture, and postprocessing algorithm, training parameters such as the learning rate and the weight decay coefficient also significantly influence network performance. YOLOv5 introduces a method for hyperparameter optimization known as hyperparameter evolution. This technique, leveraging genetic algorithms (GAs), enables the selection of more appropriate hyperparameters. Default hyperparameters were determined through hyperparameter evolution on the COCO dataset. This study employs hyperparameter evolution to identify network parameter settings optimized for our dataset, resulting in a marginal efficiency improvement. The hyperparameter adjustments are detailed in Table 4. Due to the extensive array of hyperparameters involved in the training configuration, this document only presents those that were altered.

## 5. Experimental Results

This section will discuss the experimental outcomes for various enhancement techniques proposed in this study. The experiments were conducted using an i9-7920X CPU, an RTX6000 GPU, and a software environment comprising Python 3.8, OpenCV 3.4.2, CUDA 10.1, CUDNN 7.4, PyTorch 1.7.1, and TensorRT 7.0.

The thermal-image water object dataset was split into training and testing sets with an 80:20 ratio, resulting in 4589 images for training and 1147 for testing. We set the training to 300 epochs and monitored the loss value of the training set, employing early stopping to ensure the network was adequately trained to convergence when changes became negligible over an extended period.

### 5.1. Test Results of Data Augmentation

Building on the previous discussion, the enhancements applied to data augmentation included the generation of thermal image samples using CycleGAN, along with color gamut transformation. The testing methodology used is as follows: first, augment the data; then, evaluate the training sets by employing the original YOLOv5s network to assess the performance enhancements attributed to different data augmentation methods on the network.

The impact of each enhancement algorithm is summarized in Table 5. Without the data augmentation improvements, the original model’s mean average precision (mAP) stands at 83.5%. The utilization of CycleGAN for generating thermal image samples marginally increased the model’s average precision to 83.8%, indicating a subtle enhancement effect. This data augmentation strategy primarily aims to expand the dataset. A significant improvement in model accuracy might be anticipated due to the generation of a substantial number of clear thermal images from the visible-light images. However, given that visible-light images possess higher spatial resolution, more vibrant color information, and clearer texture features compared to thermal images, efficiently generating clear thermal images from visible-light images presents a persistent challenge. Implementing the color gamut transformation data enhancement method raised the model’s mean average precision to 84.3%. Employing both data augmentation strategies concurrently slightly improved the mean average precision of the model to 84.4%.

### 5.2. Hyperparameter Evolution

The improvements of the network structure included two aspects: adding the CBAM to the backbone of the network to improve the feature extraction capability of the model and using Ghost convolution to reduce the amount of computation and improve the inference speed. The method used for this part is as follows: First, augment the data on the training sets by testing the test sets with the original YOLOv5s model. Then, test the performance improvement of each improved method and combine the two methods to test them together. Finally, test the performance of the improved model after data augmentation.

The results of these tests are shown in Table 6.

Incorporating Ghost convolution into the model resulted in a modest improvement in inference speed and a reduction in model size, albeit with a longer training duration. The addition of the CBAM effectively enhanced the model accuracy, though it did incur a decrease in the inference speed. The combined application of both enhancements yielded a slight increase in inference speed compared to the addition of the CBAM alone, yet the training time was further extended. Nonetheless, when contrasted with the 81.2 MB YOLOv5m network, the enhanced YOLOv5s model proposed in this study is considerably more compact, at just 13 MB, while maintaining comparable or even slightly superior performance. Consequently, this model was selected for use in our detection tasks. Evaluation of the model on the test sets utilized the mean average precision (mAP) for each category and a confusion matrix, as depicted in Figure 10. The confusion matrix indicates generally strong model performance, with limited misclassifications between categories and the background. The accuracy for most categories exceeds 80% according to the mAP data, with the detection of drowning individuals—critical for intelligent lifesaving missions—being notably effective. Given that individuals drowning tend to be partially submerged and can be considered small objects from a distance, achieving an initial category mAP of 80% underscores the proposed model’s strengths. The lower mAP observed for individuals on board can be attributed to the smaller sample size for this category and the fact that, during data collection, most vessels were positioned mid-river or lake, over 150 m from the shore, leading to the objects on board being significantly small. Future studies will aim to enrich the dataset with additional images of individuals on board.

### 5.3. Test Results of Training Parameter Optimization

This study leveraged the hyperparameter evolution method available within the YOLOv5 project to fine-tune the training parameters. Utilizing the enhanced YOLOv5s network, which demonstrates optimal performance with default training parameters, as a pre-trained model, we conducted a 300-epoch training session for hyperparameter evolution, yielding a new set of training parameters. The network was then trained using both the default and the newly optimized hyperparameters. The test outcomes are presented in Table 7. While the optimization of hyperparameters resulted in a reduction in model accuracy, it also decreased the training duration. The default parameters, derived from training on the COCO dataset, may not align perfectly with our dataset, which significantly differs in scale. This discrepancy suggests that the diminished effectiveness of hyperparameter evolution based on our dataset could stem from its limited size relative to the COCO dataset.

Considering these findings, revisiting this method after expanding the dataset in future studies could be beneficial.

### 5.4. Display of Test Results

The final approach involved incorporating the thermal image generation and color gamut transformation methods based on CycleGAN for data augmentation; utilizing the Ghost convolution module to substitute the standard convolution module in the YOLOv5s network; integrating the CBAM to enhance the network performance; and employing the default training parameters for network training.

A schematic representation of selected detection results from the enhanced YOLOv5s network on the test datasets is depicted in Figure 11. The results demonstrate the detection performance for different activities under various environmental temperatures.

## 6. Conclusions and Future Prospects

In addressing the challenges of intelligent tasks for lifesaving in water, this paper explores a method for detecting water objects at night using thermal imaging technology. Initially, we construct a dataset of thermal images of water objects through field collections and model generations. Subsequently, we evaluate several optimization strategies for the YOLOv5s algorithm:(1)Enhancing data augmentation by leveraging the CycleGAN technique to generate thermal images from visible-light images and applying color gamut transformation for more effective augmentation tailored to thermal images in aquatic environments.(2)Refining the YOLOv5s network structure by incorporating Ghost convolution to reduce computational demands and integrating the CBAM to bolster the model’s feature extraction capabilities.(3)Tailoring the training parameters for the thermal-image water object dataset using hyperparameter evolution.

The method we propose possesses the following characteristics:(1)We developed a dataset covering different environmental conditions by utilizing color gamut transformation and CycleGAN enhancement techniques.(2)The integration of the Convolutional Block Attention Module (CBAM) enhances the network’s capability to capture details in feature maps.(3)The employment of Ghost convolution significantly improves the model’s inference speed.(4)The adoption of hyperparameter evolution method notably enhances both the accuracy and inference speed of the model.(5)The experimental findings demonstrate that the optimized YOLOv5s network performs commendably on our custom thermal-image water object dataset, showcasing stability, speed, and accuracy.

The field of water object detection using thermal imagery is burgeoning with potential. Our research represents an initial foray into this domain, with further exploration required:(1)Aquatic environments are highly variable, with significant differences across rivers, lakes, and oceans, making dataset expansion in these settings a substantial challenge. Additionally, collecting thermal images of actual drowning incidents poses a significant hurdle.(2)The universality of the model is crucial. The dataset used for training and testing in this paper is limited in terms of environmental conditions, and it lacks specific categorization according to environmental conditions. This led to a lack of comparative experiments in different environmental conditions in this paper, which affects the effectiveness of the model in practical applications. Therefore, it is essential to classify and collect datasets from different environmental conditions.(3)For unmanned lifesaving missions, this study categorized various states of individuals in water, such as swimming and drowning. However, the dynamic nature of swimming and drowning movements, which unfolds over time, cannot be fully captured through single-frame images. Investigating how to leverage dynamic temporal information from sequential video frames to enhance human object recognition presents a promising avenue for improvement [12].

## Figures and Tables

**Figure 1 sensors-24-03321-f001:**
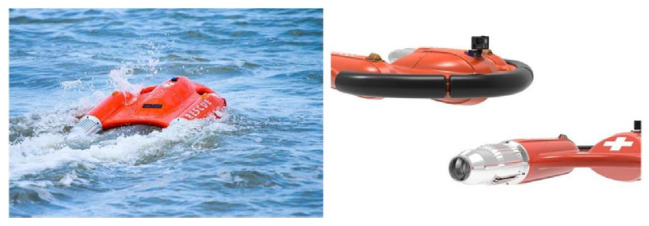
“Dolphin I” water lifesaving robot.

**Figure 2 sensors-24-03321-f002:**
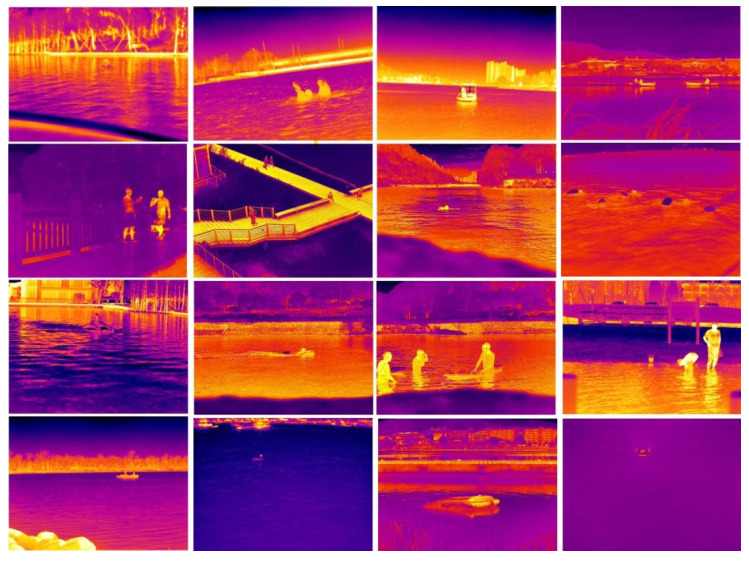
Display of the IR-YZ dataset.

**Figure 3 sensors-24-03321-f003:**
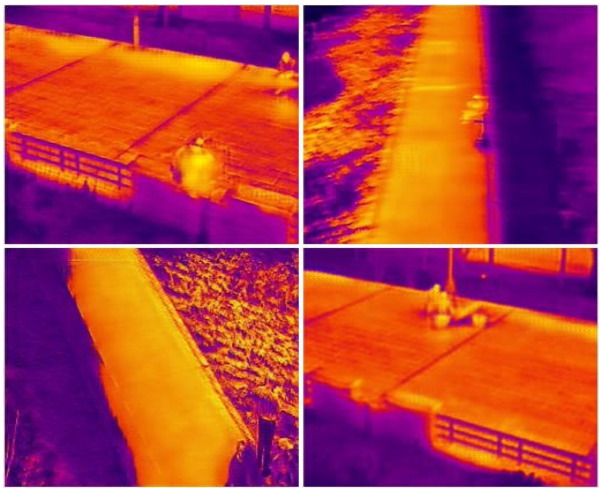
Thermal images generated from CycleGAN.

**Figure 4 sensors-24-03321-f004:**
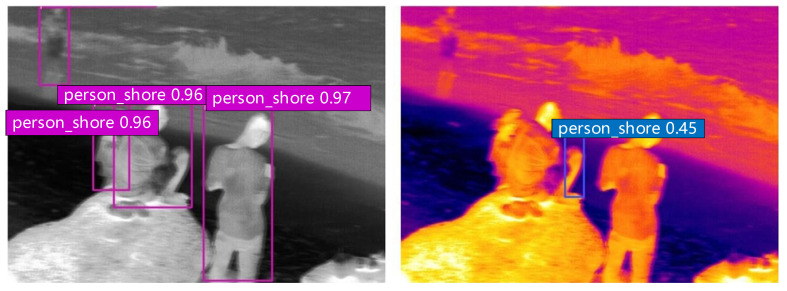
Comparison of the detection of the model on thermal images with different color-toning methods.

**Figure 5 sensors-24-03321-f005:**
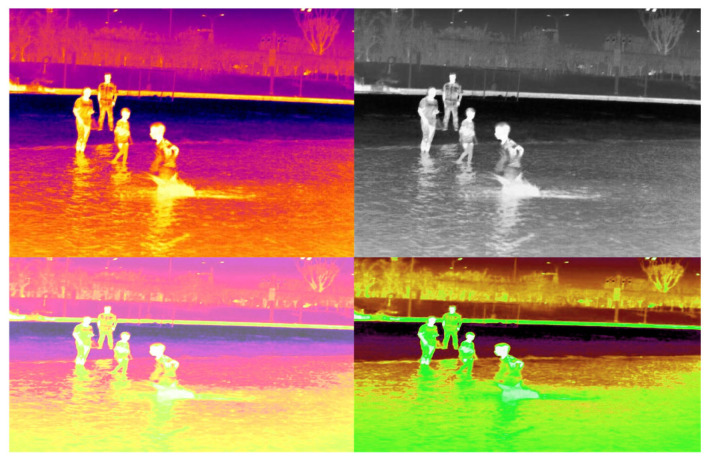
Gamut transformation data enhancement.

**Figure 6 sensors-24-03321-f006:**
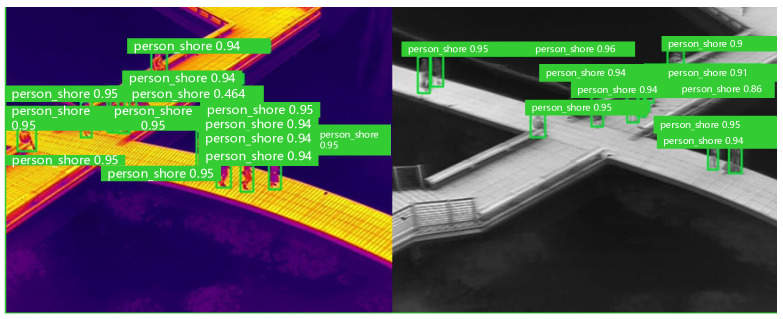
Comparisons of detection results before and after gamut transformation.

**Figure 7 sensors-24-03321-f007:**
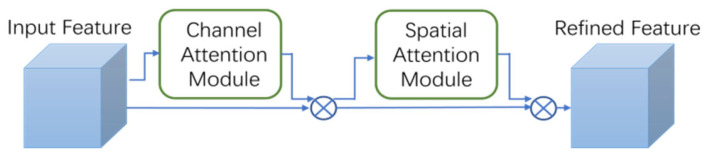
The overview of the CBAM.

**Figure 8 sensors-24-03321-f008:**
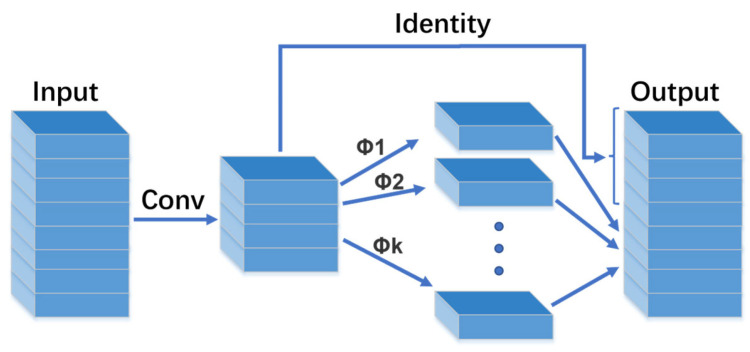
Schematic diagram of Ghost convolution.

**Figure 9 sensors-24-03321-f009:**
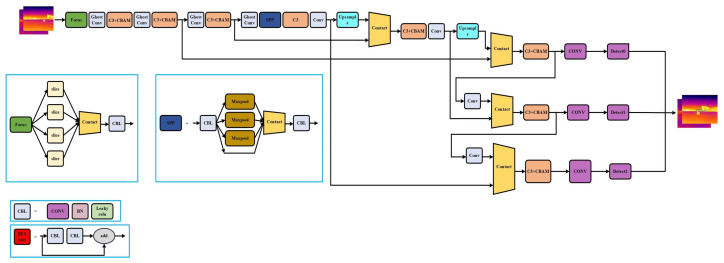
Improved YOLOv5s network structure.

**Figure 10 sensors-24-03321-f010:**
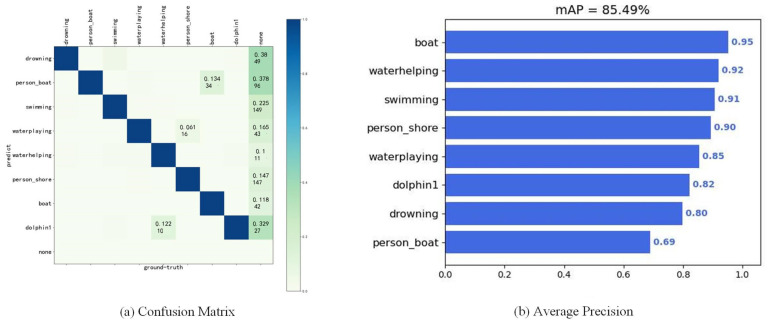
Model evolution results. (**a**) Confusion matrix. (**b**) Mean average precision of eight labels.

**Figure 11 sensors-24-03321-f011:**
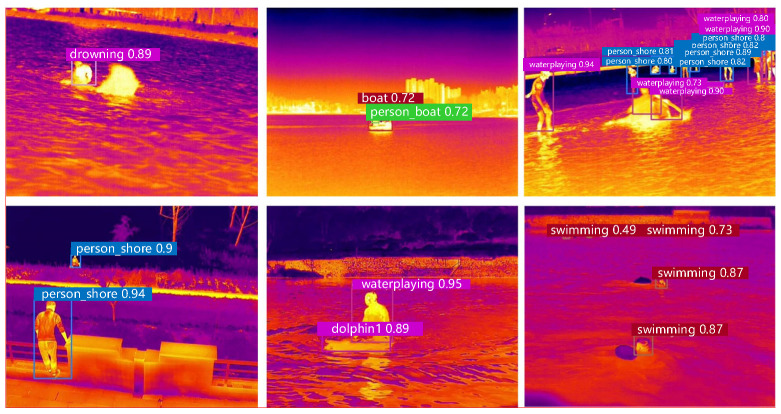
Test results.

**Table 1 sensors-24-03321-t001:** Indicators of the data acquisition instruments.

Class	MAG62	H20T
Detector type	uncooled focal plane	uncooled focal plane
Wavelength range	7.5~14 μm	8~14 μm
Pixel count	640 × 480	640 × 512
Pixel size	17 μm	12 μm
Frame rate	50 Hz	30 Hz
Operating temperature	−20~50 °C	−20~50 °C
Palette	iron red, black and white, etc.	iron red, black and white, etc.

**Table 2 sensors-24-03321-t002:** IR-YZ dataset’s composition.

Class	Number of Pictures	Number of Labels
drowning	1011	1021
person_boat	506	1696
person_shore	1530	5426
swimming	2153	3801
waterhelping	639	640
waterplaying	682	1599
boat	841	1701
dolphin1	586	589

**Table 3 sensors-24-03321-t003:** Performance comparison between mainstream object detection algorithms.

Class	mAP (%)	FPS	Time Consumption	Weight
YOLOv3	83.03	21.56	13.01 h	123.5 MB
SSD	82.92	21.25	14.12 h	178 MB
YOLOv5m	85.24	23.07	8.2 h	81.2 MB
YOLOv5s	83.51	26.14	5.4 h	14.4 MB
YOLOv5l	86.18	21.34	10 h	93.8 MB

**Table 4 sensors-24-03321-t004:** Hyperparameter optimization results.

Parameter Name	Before Optimization	After Optimization
initial learning rate	0.01	0.00816
final OneCycleLR learning	0.2	0.25725
SGD momentum	0.937	0.98
warmup_bias_lr	0.1	0.11521
image HSV-Hue augmentation	0.015	0.01734
image HSV-Saturation augmentation	0.7	0.9
image HSV-Value augmentation	0.4	0.44819
box loss gain	0.05	0.03384
cls loss gain	0.5	0.6195
warmup_epochs	3.0	2.71044
warmup_momentum	0.8	0.66111

**Table 5 sensors-24-03321-t005:** Comparison of various enhancement algorithms.

Enhancement Method	mAP (%)
Null	83.51
Thermal image generation	83.88
Gamut transformation	84.35
Thermal image generation + gamut transformation	84.42

**Table 6 sensors-24-03321-t006:** Comparison of network structure improvement methods.

Enhancement Method	mAP (%)	FPS	Model Size (MB)	Time Consumption (h)
YOLOv5s	84.42	26.14	14.4	5.40
YOLOv5s + GhostConv	84.40	26.37	12	4.36
YOLOv5s + CBAM	85.48	23.42	14.6	5.74
YOLOv5s + GhostConv + CBAM	85.49	23.51	13.0	6.17

**Table 7 sensors-24-03321-t007:** Comparison before and after hyperparameter optimization.

	mAP (%)	Time Consumption (h)
Default hyperparameters	85.49	6.17
Improved hyperparameters	84.29	6.00

## Data Availability

The data presented in this study are available on request from the corresponding author.

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
