# Peer review of "Personnel Detection in Dark Aquatic Environments Based on Infrared Thermal Imaging Technology and an Improved YOLOv5s Model"

_sensors, 2024, doi:10.3390/s24113321_

Round 1
Reviewer 1 Report
Comments and Suggestions for Authors
1. The introduction of Part 1 presents four issues, with explanations provided for the solutions to these issues. However, for some issues, only the results are presented without the experimental process and resolution methods, such as the impact of environmental temperature or weather conditions on the experiments.
2. It is recommended to test the generalized capabilities of the improved YOLO model under various environmental conditions, such as different water qualities, weather, and lighting conditions. This could be achieved by conducting more field experiments or using simulated environments to ensure that the model not only performs well on specific datasets but also maintains high accuracy and robustness under the diverse conditions of the real world.
Comments on the Quality of English Language1. In Section 5, two paragraphs are completely repeated:
“This section will discuss the experimental outcomes for various enhancement techniques proposed in this study. The experiments were conducted using an i9-7920X CPU, an RTX6000 GPU, and a software environment comprising Python 3.8, OpenCV 3.4.2, CUDA 10.1, CUDNN 7.4, PyTorch 1.7.1, and TensorRT 7.0.”
2. There are two subsections labeled 5.1.
Reviewer 2 Report
Comments and Suggestions for Authors
Title and content of abstract of this paper are promising, however some important features of the proposed advanced method not described and some of expected results not presented in the paper.
It’s necessary to describe and show samples of thermal images how the developed method works in such different conditions:
11. When the environmental temperature approximates that of the objects, for example, air temperature same as temperature of human body.
22. When it’s rainy.
33. When it’s windy and there are significant waves on the water.
Round 2
Reviewer 2 Report
Comments and Suggestions for Authors
This paper presents a novel method for human target detection technology in nighttime aquatic environments. It’s shown that the method works in good weather and still water conditions and has a potential for further development to be applicable in different weather and water waves conditions.